# Effects of Simulated Microgravity In Vitro on Human Metaphase II Oocytes: An Electron Microscopy-Based Study

**DOI:** 10.3390/cells12101346

**Published:** 2023-05-09

**Authors:** Selenia Miglietta, Loredana Cristiano, Maria Salomé B. Espinola, Maria Grazia Masiello, Giulietta Micara, Ezio Battaglione, Antonella Linari, Maria Grazia Palmerini, Giuseppe Familiari, Cesare Aragona, Mariano Bizzarri, Guido Macchiarelli, Stefania A. Nottola

**Affiliations:** 1Department of Anatomy, Histology, Forensic Medicine and Orthopaedics, Sapienza University, 00165 Rome, Italy; 2Department of Life, Health and Environmental Sciences, University of L’Aquila, 67100 L’Aquila, Italy; 3Systems Biology Group, Department of Experimental Medicine, Sapienza University, 00165 Rome, Italy; 4Department of Maternal, Infantile and Urological Sciences, Sapienza University, 00165 Rome, Italy

**Keywords:** simulated microgravity, electron microscopy, oocyte, organelles, ultrastructure, human

## Abstract

**Highlights:**

**What are the main findings?**
For the first time, the in vitro effects of simulated microgravity using a Random Positioning Machine on the ultrastructural features of human metaphase II oocytes were investigated by transmission electron microscopy.Microgravity might compromise oocyte quality affecting the ultrastructural morphology of mitochondria, endoplasmic reticulum, and cortical granules due to a possible alteration of the cytoskeleton.

**What is the implication of the main finding?**
Microgravity-induced organelle modifications potentially affect the oocyte’s ability to fully mature and fertilize and develop into a viable embryo.

**Abstract:**

The Gravity Force to which living beings are subjected on Earth rules the functionality of most biological processes in many tissues. It has been reported that a situation of Microgravity (such as that occurring in space) causes negative effects on living beings. Astronauts returning from space shuttle missions or from the International Space Station have been diagnosed with various health problems, such as bone demineralization, muscle atrophy, cardiovascular deconditioning, and vestibular and sensory imbalance, including impaired visual acuity, altered metabolic and nutritional status, and immune system dysregulation. Microgravity has profound effects also on reproductive functions. Female astronauts, in fact, suppress their cycles during space travels, and effects at the cellular level in the early embryo development and on female gamete maturation have also been observed. The opportunities to use space flights to study the effects of gravity variations are limited because of the high costs and lack of repeatability of the experiments. For these reasons, the use of microgravity simulators for studying, at the cellular level, the effects, such as those, obtained during/after a spatial trip, are developed to confirm that these models can be used in the study of body responses under conditions different from those found in a unitary Gravity environment (1 g). In view of this, this study aimed to investigate in vitro the effects of simulated microgravity on the ultrastructural features of human metaphase II oocytes using a Random Positioning Machine (RPM). We demonstrated for the first time, by Transmission Electron Microscopy analysis, that microgravity might compromise oocyte quality by affecting not only the localization of mitochondria and cortical granules due to a possible alteration of the cytoskeleton but also the function of mitochondria and endoplasmic reticulum since in RPM oocytes we observed a switch in the morphology of smooth endoplasmic reticulum (SER) and associated mitochondria from mitochondria-SER aggregates to mitochondria–vesicle complexes. We concluded that microgravity might negatively affect oocyte quality by interfering in vitro with the normal sequence of morphodynamic events essential for acquiring and maintaining a proper competence to fertilization in human oocytes.

## 1. Introduction

The biological effects of microgravity exposure on the human body have been recognized and investigated since the earliest space flights, forming one of the most important research directions in the field of aerospace medicine [1]. Exposure to microgravity during space flight induces in astronauts several significant physiological changes, including a loss of muscle mass and bone density, impaired vision, altered kidney function, diminished neurological response, and a compromised immune system, venous system, and blood coagulation [1,2,3,4]. Recent studies highlight the impact of microgravity also at a cellular level. Many data show that microgravity alters, permanently or temporarily, important biological processes, such as differentiation, maturation degrees, survival, and cell morphology [5]. The cytoskeleton, which plays fundamental roles in mitosis, maintenance of cell shape, cell motility, and cell adhesion [6], seems particularly sensitive to gravity [7]. Alteration of the cytoskeleton observed under simulated microgravity induced relocation and clustering of cytoplasmic organelles, such as mitochondria [8].

Microgravity may impair mitochondrial energy potential [9,10] and affect mitochondrial function in all cell types by increasing mitochondrial ROS levels, thus inducing autophagy [11]. Even microgravity-induced cytoskeletal impairment may be directly connected to autophagy [12].

Regarding the effects of microgravity on reproductive functions, it is known that the stress suffered during spaceflights may be linked to detrimental effects on the ovulatory cycle; in fact, female astronauts suppress their cycles [13]. At the cellular level, microgravity influences early embryo development [14] and has a role in the modulation of the dynamic events correlated to the acquisition of developmental competence in the oocyte; microfilaments and microtubules, as cytoskeletal components, play, in fact, a crucial role driving cortical granules (CGs) and mitochondria movements during oocyte maturation [15,16]. Indeed, cytoskeletal alterations have been associated with dysmorphic patterns in human oocytes [17].

Since the opportunities to use space flights to study the effects of gravity variations are limited because of the high costs and lack of repeatability of the experiments (Goswami et al., 2021) [18] and in view of the importance of cytoskeleton and mitochondrial function in the normal physiology of the human oocyte, we analyzed the microanatomical ultrastructural characteristics of human oocytes subjected to simulated microgravity by random positioning machine (RPM). Clinorotation was the first method designed to simulate microgravity on the ground, and it remains the most common and accessible simulation procedure because it allows the rotation of biological samples along two independent axes to change their orientation in space, thus eliminating the effect of gravity [19,20]. Effects generated by the RPM are comparable to those of real microgravity, provided that the direction changes are faster than the response time of the system to the gravity field. Therefore, RPM is a laboratory instrument that can simulate some aspects of microgravity and a weightless environment during spaceflight with good approximation to the microgravity environment on Earth.

The objective of the present investigation was, thus, to examine for the first time by Light and Transmission Electron Microscopy (LM and TEM) the morphological features of human metaphase II (MII) oocytes subjected to simulated microgravity for 24 h (RPM oocytes) compared to those shown by human MII oocytes cultured on the ground (control oocytes), to reveal eventual microanatomical ultrastructural changes in oocyte morphology and organelle morphodynamics possibly induced by microgravity.

## 2. Materials and Methods

### 2.1. Oocyte Recovery

The oocytes were obtained from 13 patients aged 26 to 35 years (mean age 31.2 ± 3.14) undergoing ART treatments performed at the Infertility and Assisted Reproduction Unit, Department of Gynecology–Obstetrics and Urology (now Department of Maternal, Infantile, and Urological Sciences), “Sapienza” University of Rome, Italy, from November 2016 to June 2017. All patients included in this study had a normal karyotype and normal clinical and hormonal assessments.

The study was conducted in accordance with the Declaration of Helsinki and approved by the Ethics Committee (Azienda Policlinico Umberto I, Sapienza University of Rome, Italy; reference number 3558, protocol number 1963/15, 31 March 2015) for studies involving humans, and upon enrolment all participants were required to sign an informed consent.

IVF/ICSI cycle management consisted of downregulation with a long protocol starting from day 21 of the pre-treatment cycle with a GnRH agonist (Decapeptyl 0.1 mL, IPSEN/BIOTECH, Boulogne-Billancourt, France). Once ovarian suppression was assessed by 17ß-estradiol(E2) profiles and ovarian ultrasound scan (US), daily s.c. administration of 300 IU of recombinant FSH (Gonal F, Serono^®^, Geneva, Switzerland) was commenced.

From the fifth day of stimulation, daily monitoring of follicles size by the US was performed, and plasma levels of E2 and progesterone were measured. From this stage, the dose of gonadotropins was adjusted depending on the individual response of each patient. Criteria used for triggering ovulation with 10,000 IU hCG (Gonasi HP 10,000, Ibsa^®^, Lugano, Switzerland) s.c. were plasma E2 between 1000 and 3000 pg/mL and at least four follicles > 18 mm mean diameter (two perpendicular measurements) with plasma progesterone < 1.5 ng/mL.

Oocyte retrieval was performed 36 h after hCG administration by transvaginal US-guided follicular aspiration under intravenous sedation. The selection of samples for our study was performed according to the Italian law regulating ART at the time of the pickup (law 40/2004 and subsequent modifications).

At the time of follicular aspiration, MII oocytes were identified by a phase contrast microscope and rinsed with a buffer. Only MII oocytes that appeared of good quality when observed by PCM were included in this study. They were selected based on the following: (1) a rounded, regular shape; (2) a clear, moderately granular ooplasm; (3) a narrow perivitelline space (PVS) with the 1st polar body (PBI); and (4) an intact, colorless zona pellucida (ZP). In total, 26 oocytes were selected for Electron Microscopy (EM) analysis. Among them, 20 oocytes were subjected to simulated microgravity (RPM oocytes). The remaining 6 oocytes were assigned to the control group (control oocytes).

### 2.2. Clinorotation to Simulate Microgravity

Clinorotation was performed using the 3D-clinostat (Dutch Space, Leiden, The Netherlands), a Random Positioning Machine (RPM) capable of simulating microgravity on the ground, as described by Borst et al. (2009) [19]. The desktop RPM we used has been positioned within an incubator (for maintaining temperature, CO_2_, and humidity levels) and connected to the control console through standard electric cables.

### 2.3. Microgravity Exposure and Cell Treatments

The oocytes were cultured in multi dishes (Nunc™ 4-Well Dishes for IVF, Thermo Fischer Scientific, Waltham, MA, USA), filled with Human Tubal Fluid Modified (HTF-HEPES) supplemented with 10% of Human Serum Albumin (HSA) to avoid the presence of air bubble, capped, and transferred into a Desktop RPM. In the experimental conditions of our study, oocytes were exposed continuously in the RPM for 24 h (RPM oocytes). Similarly, control oocytes were maintained for an analogous period in the same HTF-HEPES medium and 10% of HSA and cultured on ground (static) conditions. After 24 h, all the oocytes (both RPM and control oocytes) were recovered, immediately photographed, and fixed for further morphological structural and ultrastructural evaluation.

### 2.4. Light/Electron Microscopy

The oocytes were prepared for EM observations according to Coticchio et al. (2016) [21]. They were fixed with 2.5% glutaraldehyde (Electron Microscopy Sciences, Hatfield, PA, USA) in phosphate-buffered saline solution (PBS) for at least 48 h at 4 °C. Fixed samples, after several items of washing in PBS, were post-fixed with a 1% osmium tetroxide (Electron Microscopy Sciences) for 2 h, rinsed several times in PBS and embedded in agar 1% (Electron Microscopy Sciences). Oocytes were then rinsed in PBS, dehydrated through ascending series of ethanol, immersed firstly in propylene oxide (Sigma-Aldrich. St. Louis, MO, USA) for 40 min, and then left overnight in a propylene oxide/resin 1:1 solution. Finally, they were embedded in epoxy resin (Agar Scientific, Stansted, UK) for 48 h at 60 °C. Resin blocks were sectioned using an Ultracut E ultramicrotome (Leica EMUC6, Wetzlar, Germany).

For LM observations (Zeiss Axioscope 40, Gottingen, Germany), semithin sections (1 μm) were colored with Azur II staining (Sigma-Aldrich); for TEM analyses (Zeiss EM10, Oberkochen, Germany), operating at 60 kV, ultrathin sections (90–100 nm) were cut with a diamond knife, mounted on 100-mesh copper grids (Assing, Rome, Italy), contrasted with Uranyless (Uranyl acetate alternative) (TAAB Laboratories Equipment Ltd., Alder-maston, UK) and lead citrate (Electron Microscopy Sciences)and photographed. Images were acquired using a digital camera (AMT CCD, Deben UK Ltd., Suffolk, UK).

### 2.5. Statistical Analysis

The evaluation of the number of mitochondria, mitochondria-smooth endoplasmic reticulum (M-SER) aggregates, mitochondria–vesicle (MV) complexes, and CGs was performed through the collection of TEM microphotographs at 6300× magnification on three equatorial sections per oocyte, according to Coticchio et al. (2016) [21]. Values were expressed as the number of M-SER and MV per 100 µm^2^ of oocyte area and as the number of CG for 10 µm of the oocyte linear surface profile. Data were expressed as mean ± standard deviation and compared by one-way analysis of variance (ANOVA). Differences in values were considered significant at *p* < 0.05. Cumulative distribution functions were compared by the Kolmogorov–Smirnov test using the Origin software.

## 3. Results

### 3.1. Sample Characteristics by Light Microscope

Figure 1 shows representative equatorial semithin sections of control and RPM human mature (MII) oocytes by LM. In control oocytes (Figure 1a,c), the shape of the oocyte is rounded, the PVS is narrow and uniform, the ZP—approximately 10–12 μm thick—is intact, organelles appear uniformly distributed in the ooplasm, and the CGs are abundant and regularly aligned under the oolemma; in RPM oocytes (Figure 1b,d) the oocyte maintains its rounded contour (only occasionally the shape of RPM oocytes displayed a slight distortion, data not shown), the PVS is unevenly expanded, the ZP thickness appears increased (>12 μm), organelles are slightly concentrated in the inner areas of the ooplasm, and the CGs do not appear regularly aligned under the oolemma, providing evidence of a possible metabolic distress.

### 3.2. Ultrastructural Features in the Ooplasmic Compartment

The ultrastructural analysis of ultrathin sections of control and RPM human MII oocytes (Figure 2) by TEM highlights that associations between mitochondria and SER elements are abundant in the ooplasm of both groups. However, in control oocytes (Figure 2a,b), typical associations between mitochondria and tubular elements of SER (M-SER aggregates) are observable and interspersed among numerous MV complexes of various sizes, including unusually large complexes. On the contrary, in RPM oocytes (Figure 2c,d), different from what is observed in control oocytes (Figure 2a,b), the amount of M-SER aggregates decreases, whereas large MV complexes increase in number, reversing the ratio of the two structures, and thus indicating that microgravity could affect endoplasmic reticulum (ER) shape.

The morphometric analysis of the number of M-SER aggregates and MV complexes on a total of 26 oocytes (6 control oocytes and 20 RPM oocytes) (Figure 3a) confirms the TEM observations and suggests that after the exposure to microgravity oocytes are subjected to metabolic stress, which determines the switch from M-SER aggregates to MV complexes. In fact, in RPM oocytes, M-SER aggregates significantly diminished; on the contrary, MV complexes significantly augmented in number. Figure 3b highlights that all RPM oocytes exhibit this phenotype. Figure 3c,d show the cumulative distribution functions of M-SER and MV analyzed by the Kolmogorov–Smirnov test, demonstrating that both organelles exhibit significantly different distribution in control and RPM oocytes.

The results obtained by morphometric analysis of the number of mitochondria per 100 µm^2^ of oocyte area demonstrated that there are no significant changes between the two experimental groups (control and RPM oocytes) (Figure 4a,b). Moreover, the analysis by the Kolmogorov–Smirnov test (Figure 4c) indicates that the cumulative distributions of organelles are not significantly different between control and RPM oocytes.

By TEM, mitochondria generally appear rounded or oval and are provided with peripheral, arched cristae in both groups of oocytes (Figure 2 and Figure 5a). Interestingly, in some RPM oocytes (40%), several mitochondria associated with vesicles begin to stretch around them, sometimes appearing elongated, hooded, or showing a dumbbell shape, and are thought to divide, as indicated in Figure 5b–f, in a percentage of 50% of the total mitochondrial number. On the contrary, this type of mitochondria (in fission) has been only occasionally found in the analyzed sections of control oocytes.

Since microgravity affects the ER and ER, together with the actin cytoskeleton, are also involved in the mitochondrial division, we assume that in response to a stress stimulus induced by simulated microgravity, mitochondria may undergo morphodynamic changes, closely correlated to the cell metabolism. High levels of cell stress may lead to excessive fission of mitochondria. Moreover, mitochondrial structural remodeling through division and fusion is critical to the organelle’s function.

Besides the increase in large MV complexes and the presence of elongated, hooded, or dumbbell-shaped, thus possibly dividing, mitochondria in RPM oocytes, mitochondrial clustering also appears in 20% of oocytes belonging to this group. Unlike controls, in these latter RPM oocytes, mitochondria are detached from ER vesicles and gather, forming large clusters in specific areas of the ooplasm, as shown in Figure 6. Even in this case, a correlation between this morphodynamic feature and an alteration of the cytoskeleton induced in cells exposed to simulated microgravity should not be excluded.

Mitochondria with altered morphology can also be found in some areas of RPM oocytes, as shown in the representative images of Figure 7. As shown in Figure 7a, the mitochondria of control oocytes are mainly spherical/oval with peripheral, arched, or transverse cristae and a diameter of 0.4–0.8 µm, all typical morphological features for the mitochondria of human oocytes. In contrast, mitochondria of some RPM oocytes (50%), in a percentage of 30% of the mitochondrial total number, display an irregular contour and/or appear swollen (with a diameter larger than 0.8 µm), with collapsed cristae and a denser matrix. Some of them appear vacuolated or include debris (Figure 7b–d). Altered mitochondria have been found only occasionally in control oocytes.

Mitochondria, therefore, appear to be variously influenced by microgravity conditions not only as fundamental components of M-SER aggregates and MV complexes but also because they are presumably induced to divide, group, and/or regress by microgravity, leading us to speculate a correlation between such peculiar mitochondrial morphodynamic changes and the occurrence of various mitochondrial dysfunctions during and after cell exposure to microgravity.

The hypothesis of a possible alteration of the oocyte cytoskeleton as an effect of microgravity seems sustained by the ultrastructural observation of an altered distribution of the CGs in RPM oocytes (Figure 8). In control oocytes, a continuous rim of electron-dense cortical granules is seen slightly beneath the oolemma (Figure 8a). Differently from the controls, RPM oocytes exhibit a reduced amount of CGs under the oolemma, forming a discontinuous rim, whereas several CGs are instead found deeply in the ooplasm, showing a sort of retrograde migration (relocation), thus indicating the occurrence of a possible alteration of actin microfilaments, that drive proper CG migration and positioning (Figure 8b). Morphometric data do not confirm these ultrastructural differences among RPM and control oocytes, revealing that microgravity does not significantly affect CG number in RPM oocytes (Figure 8c,d), although the cumulative distributions of organelles are significantly different between control and RPM oocytes, as demonstrated by the Kolmogorov–Smirnov analysis (Figure 8e).

Figure 9 shows the presence of vacuoles (Figure 9a), multivesicular bodies (Figure 9b), and lysosomes (Figure 9c,d) in the ooplasm of RPM oocytes. Associations among multiple vacuoles, multivesicular bodies, and lysosomes were found in a percentage of RPM oocytes (50%). On the contrary, only occasional, isolated vacuoles were sometimes found in controls. The presence of multiple vacuoles associated with swelling and coalescence of SER vesicles, multivesicular bodies, and lysosomes indicates that microgravity could induce metabolic suffering leading to a cellular regression up to autophagy.

## 4. Discussion

The study aimed to investigate the effects of simulated microgravity on the ultrastructure of human MII oocytes. In fact, despite the important effects induced by the absence of gravity on human systems, organs, tissues, and the human female reproductive system in particular [22,23], only a few reports have been published about mammalian female gametes in weightlessness conditions [16,24,25]. Zhang et al. (2016) [16] evidenced that simulated microgravity affects the in vitro development of mouse preantral follicles. Wu et al. (2011) [24] demonstrated that the simulated microgravity inhibits mouse oocyte maturation via altering spindle organization and inducing cytoplasmic blebbing. Chen et al. (2023) [25] reported that simulated microgravity reduces the quality of mouse ovarian follicles and oocytes by disrupting oocyte-follicle cell intercellular contacts.

In the present study, the observation by LM of equatorial sections of RPM MII oocytes revealed in some samples modifications of the PVS that expanded asymmetrically and/or variations in the ZP thickness, which appeared increased in comparison with controls. However, the focal increase in width of the PVS, usually narrow in normal, healthy MII oocytes, could also be explained by the possible presence of the PBI in neighboring areas, not included in the section. Similarly, slight variations in ZP thickness may be due to the section plane (not exactly equatorial in all samples) [26]. Thus, we cannot attribute with certainty the occurrence of these presumptive alterations to the effects of microgravity.

A slight distortion of the oocyte shape has been only occasionally detected by LM, suggesting that simulated microgravity may grossly alter the oocyte contour due to its effects on the structure and organization of the cytoskeleton responsible for the maintenance of cell shapes [17,27]. However, this event seems reversible, and most oocytes are presumably able to recover their shape at the end of the procedure, as demonstrated by our data.

Moreover, by TEM analysis, we demonstrated, for the first time, that microgravity may compromise human oocyte quality by affecting not only the localization of mitochondria and CGs due to a possible alteration of the cytoskeleton but also the morphology of mitochondria and mitochondria-ER associations even triggering early regressive changes, possibly culminating into autophagy.

Table 1 reported the morphometric comparison of some cytoplasmic organelle distribution in control and RPM oocytes.

In detail, we observed an interesting switch in the morphology of SER membranes and associated mitochondria from M-SER aggregates to MV complexes in human MII oocytes subjected to simulated microgravity. In human mature oocytes, most mitochondria are physically associated with SER membranes to form agglomerates composed of variable elements of ER and multiple mitochondria called M-SER aggregates and MV complexes. M-SER aggregates are composed of anastomosing tubules of SER surrounded by several mitochondria and are mainly located in the cortical areas of the ooplasm; MV complexes are, instead, usually formed by small vesicles surrounded by mitochondria, appearing as “necklace-like” structures distributed in the ooplasm [26,28,29,30,31,32,33,34]. Larger, spherical sacs of ER decorated with mitochondria become, instead, more numerous in human oocytes subjected to prolonged culture (in vitro matured and in vitro aged oocytes) and in oocytes sampled from older patients (reproductive aging, patients age: ≥35 years) [21,35]. ER tubules and vesicles belong to the same system of interconnected membranes; thus, they become capable of transforming into each other under an appropriate stimulus, as our research group has previously demonstrated [36]. Such a morphodynamic process of membrane remodeling, orchestrated by the cytoskeleton, leads to the transition between M-SER aggregates and MV complexes.

Since M-SER aggregates and MV complexes act as a reserve of energy, nutrients, growth factors, membranes, and calcium, properly driving the oocyte to fertilization [37,38,39], anomalies in this organelle distribution raise concerns about the developmental competence of affected oocytes [34].

From a merely qualitative point of view, we observed that large MV complexes were present in both control and RPM oocytes as an effect of the prolonged period of culture to which both groups were subjected (24 h) [35].

However, our morphometric quantitative data demonstrated that M-SER aggregates, large and abundant in control oocytes, decrease significantly in number in RPM oocytes, whereas large MV complexes significantly increase in amount, replacing the former.

We hypothesized that the stress induced by microgravity might further induce, with an additive effect, the above-described dynamic process of transition induced by prolonged culture, compromising the effectiveness of calcium management. So, microgravity might induce a reduction in the calcium storage in ER, consequently to ER dysfunction, similar to what was observed by Nottola et al. (2016) [36] in cryopreserved human mature oocytes where freeze/thaw stress induces a membrane remodeling between M-SER aggregates and MV complexes. Microgravity may affect ER morphology, remodeling, and function by impairing the oocyte cytoskeleton [40]. Finally, since the presence of numerous, large MV complexes represents an ultrastructural marker of aging (both in vitro and in vivo aging), we cannot exclude that simulated microgravity may act as “accelerating” the aging of the oocytes in the RPM machine, mimicking what happens on human systems, organs and tissues subjected to a microgravity environment in the space and on the ground [41,42].

Our observations agree with Feger et al. (2016) [43], who confirm that microgravity in cardiomyocytes induces stress of ER, affecting protein synthesis at the proteomic level and attempting to maintain mitochondrial homeostasis at the expense of protein synthesis. Moreover, simulated microgravity also causes ER stress in human umbilical vein endothelial cells (HUVECs), subsequently inducing endothelial inflammation and apoptosis [44].

In addition to the transition of M-SER aggregates to MV complexes, we observed in RPM MII oocytes numerous elongated, hooded, or dumbbell-shaped mitochondria that are presumably dividing through fission in the proximity of ER vesicles or encircling them.

Mitochondria with an arc-like structure or “hooded” mitochondria have been described in oocytes from some species, including cows, sheep, and, more rarely, humans [45,46]. The functional significance of these mitochondrial morphologies is unclear, but the morphology may reflect changes to the energetic state of the mitochondria [46,47,48]. In particular, the presence of hooded mitochondria can be associated with specific cellular functions, such as increased oxidative capacity or playing a role in intracellular signaling pathways, including the release of mitochondrial proteins into the cytoplasm, which triggers the apoptotic process [49]. Hooded mitochondria can also be associated with oocyte aging [46]. Dumbell-shaped mitochondria are frequently seen in cells actively dividing by mitosis, thus usually considered a morphological marker of healthy, functional mitochondria. On the other hand, mitochondrial fission can also help to distribute damaged mitochondria to lysosomes for degradation, maintaining cellular homeostasis and preventing cell damage [47,48,49].

It is also known that mitochondrial structural remodeling through division and fusion is critical to the organelle’s function and that high levels of cell stress may lead to excessive fission of mitochondria. Fission is needed to create new mitochondria, but it also contributes to quality control by enabling the removal of damaged mitochondria during high levels of cellular stress [47,48,49].

Thus, mitochondrial fission may occur in response to various stimuli, including changes in energy demand or stress.

Based on the above considerations, we assume that, although the total number of mitochondria does not significantly vary in both experimental groups (control and RPM oocytes); nevertheless, in response to stress induced by simulated microgravity, mitochondria undergo dynamic morpho-functional changes, closely correlated to cell metabolism.

In RPM oocytes, we also found irregular, swollen mitochondria. Modifications of the shape of mitochondria, together with cristae alterations, such as collapse, swelling, and reduced density, have been reported in other cell types under weightlessness conditions, associated with an increase in reactive oxygen species (ROS) production and intracellular ATP levels, imbalance of mitochondrial gene expression, mitochondrial DNA damage, and autophagy. The discrepancy between ROS production and antioxidant defense in mitochondria is the main cause of mitochondrial stress and damage, which leads to mitochondrial dysfunction [11,50,51].

Mitochondrial energy potential impairment has also been demonstrated in various cells and tissues exposed to simulated microgravity, such as human primary fibroblasts [9], skeletal muscle tissue, cardiomyocytes, and plants [9,44,52,53], suggesting that mitochondrial changes could represent a common adaptative response of the cell to microgravity. However, these changes acquire special importance for the human oocyte; in fact, at fertilization, oocyte organelles (mostly mitochondria, with their genome) are inherited by the newly formed embryo, along the maternal line, by way of the oocyte cytoplasm, by-passing the Mendelian rules. Thus, the transfer of morphologically altered mitochondria from the oocyte to the embryo may even compromise early embryo development [54].

In some RPM oocytes, we found by LM an uneven distribution of the organelles in the ooplasm and, by TEM, numerous mitochondria clustered in specific areas of the ooplasm, indicating a possible alteration of microtubules in the cells exposed to simulated microgravity. Cytoskeletal and mitochondria functions are crucial for maintaining cellular homeostasis and are thought to play a role in physiological functions that are altered by spaceflight. In fact, cells subjected to microgravity may undergo cytoskeleton remodeling [5,7,55] that, in some cases, appears associated with mitochondrial modifications (changes in position and clustering), as described in cultured human lymphocytes [8].

It is worth noting that specific morphological anomalies (dysmorphisms) in human mature oocytes, associated with cytoskeletal abnormalities, can have implications for the fate of the ensuing embryo [17].

In mammalian—including human—oocytes, microfilaments, and microtubules not only drive a proper redistribution of mitochondria in the ooplasm but also mediate CG migration and destination [38]. In fact, both migration of CGs toward the periphery of the ooplasm after their production from Golgi membranes in immature oocytes and maintenance of their proper position in suboolemmal areas in MII oocytes, ready for fertilization, are events controlled by cytoskeleton [15]. Thus, our observation of an altered distribution of CGs in RPM MII oocytes could agree, again, with the assumption of a possible alteration of the cytoskeleton due to microgravity. Differently from control oocytes, where an almost continuous rim of electron-dense CGs is present slightly beneath the oolemma, RPM oocytes exhibit a few CGs below the oolemma, while several CGs apparently returned in the inner ooplasm. This suggests that microgravity may reverse CG migration, affecting CG exocytosis at fertilization, by altering actin microfilaments. However, quantitative differences in suboolemmal CG amount between control and RPM oocytes are not statistically significant.

We found the presence of associations among vacuoles, multivesicular bodies, and secondary lysosomes in the ooplasm of some RPM oocytes. Vacuolization is a form of structural damage frequently detected in human mature oocytes and derived from swelling and coalescence of isolated SER vesicles; when associated with multivesicular bodies and lysosomes, vacuoles form organelle agglomerations considered typical regressive markers for human oocytes, that may prelude to autophagy [29]. Dilatation and vesiculation of ER, associated with the uncoupling/loss of associated mitochondria, have also been observed in human-aged oocytes and correlated to oxidative stress [35,56]. Simulated microgravity might, therefore, induce oxidative stress causing dysfunction of ER and mitochondria in human MII oocytes, thus leading to ultrastructural ooplasmic alterations associated with cellular regression and degeneration.

It is known that an increase in cell apoptosis is one significant consequence of the changes in cell structure and function that occur in microgravity [8]. Moreover, the cytoskeleton also plays a role in autophagy. Growing evidence supports the notion that actin dynamics play important roles throughout the various steps of autophagy, including mitophagy [12]. Our observations agree with Zhang et al. (2016) [16], suggesting that simulated microgravity promoted autophagy in mouse oocytes by producing large vacuoles in the periphery of cells and several multilamellar bodies.

We can conclude that simulated microgravity negatively affects human MII oocyte quality in vitro by interfering with the normal sequence of biomolecular changes occurring in the oocyte for acquiring proper competence to fertilization, thus producing morphologically identifiable ultrastructural alterations in different oocyte microdomains.

We also hypothesize that physiological perturbations occurring in oocytes subjected to simulated microgravity may be a consequence of alterations in molecular and organellar dynamics. For the most part, organelle modifications may be due to alterations of the oocyte cytoskeletal scaffolding as a primary effect of microgravity. These alterations potentially affect the oocyte’s ability to fully mature, fertilize and develop into a viable embryo.

We hope that this study will improve the knowledge of the effects of microgravity on human oocyte morphology and competence since relatively little is known about the effect of spaceflights on human female reproduction. More generally, our results could contribute to the creation of a database of information that can be useful in the future to contrast the negative effects of microgravity on biological tissues, improving the quality of life of astronauts in space and upon their return to Earth.

## Figures and Tables

**Figure 1 cells-12-01346-f001:**
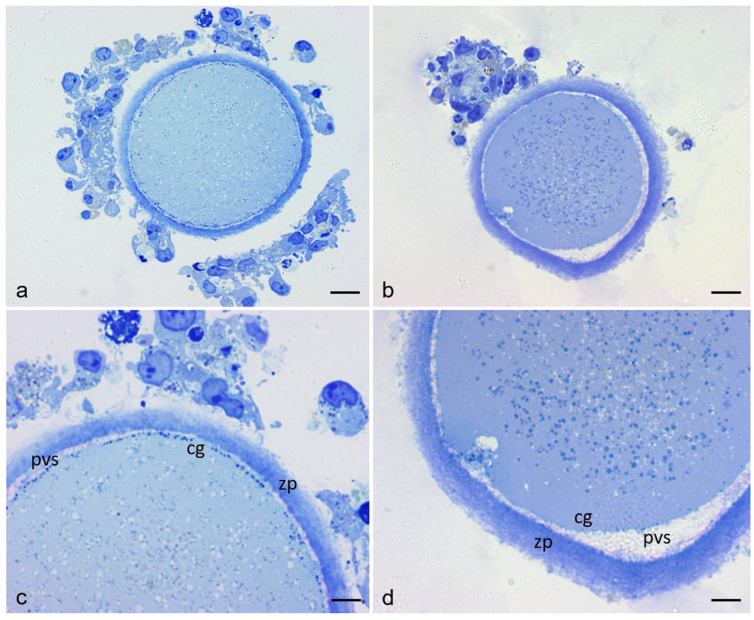
Control and RPM oocytes stained with Azur II and observed by LM. (**a**,**c**) control oocytes. (**b**,**d**) RPM oocytes. Microgravity-induced modifications include asymmetric expansion of the perivitelline space (pvs), altered localization of the cortical granules (cg) under the oolemma, uneven distribution of the organelles in the ooplasm and increased thickening of the zona pellucida (zp). Clumps of residual cumulus-corona cells are also seen around both control and RPM oocytes. Bar = 14 µm (**a**,**b**); Bar = 7 µm (**c**,**d**).

**Figure 2 cells-12-01346-f002:**
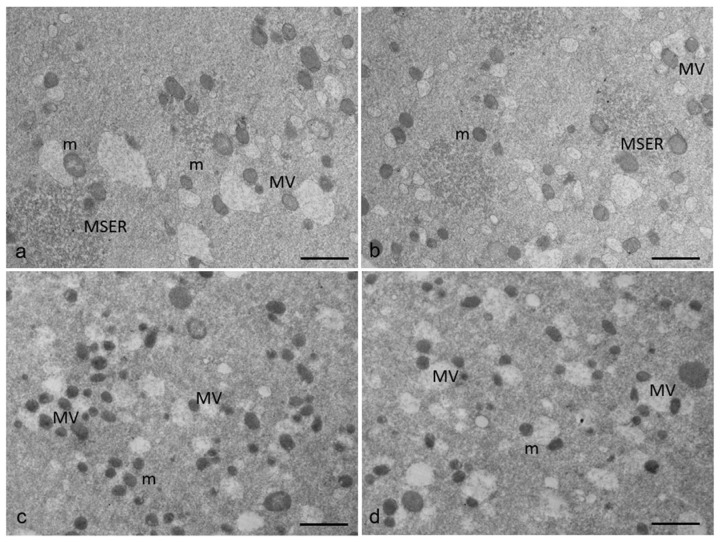
A representative picture of M-SER aggregates and MV complexes in human MII oocytes by TEM. (**a**,**b**) control oocytes. (**c**,**d**) RPM oocytes. In control oocytes, mitochondria (m) are associated with the endoplasmic reticulum to form typical large, abundant M-SER aggregates and numerous MV complexes of variable dimensions; in RPM oocytes, M-SER aggregates decrease in size and number after exposure to microgravity, whereas the opposite trend is observed for MV complexes. Bar = 1 µm.

**Figure 3 cells-12-01346-f003:**
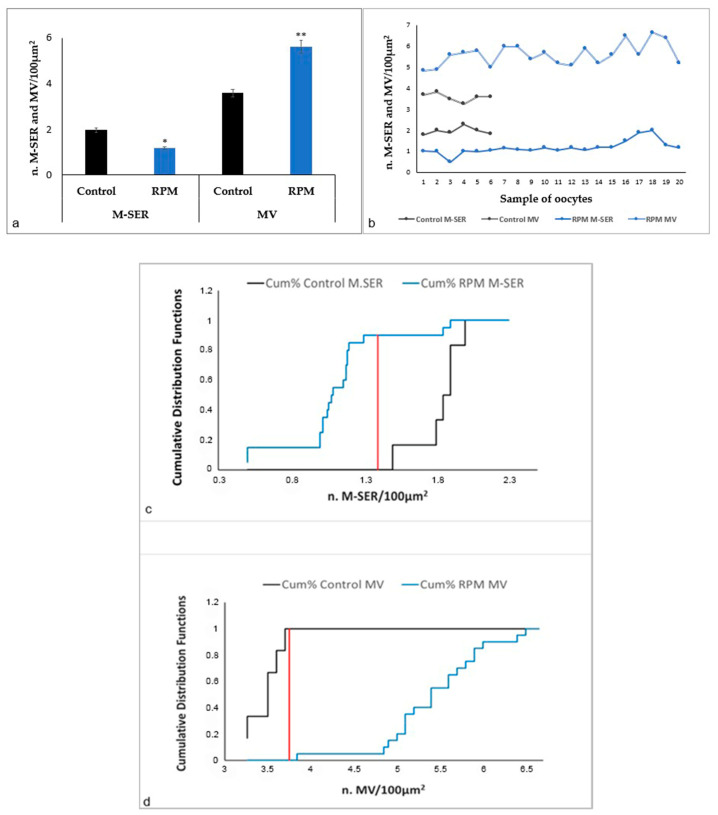
Quantitative evaluation of M-SER and MV in human MII oocytes by morphometric analysis. (**a**) Simulated microgravity significantly affects the number of M-SER aggregates and MV complexes. Values are mean ± SD of the number of M-SER aggregates and MV complexes per 100 µm^2^ of oocyte area of a total of 26 oocytes (6 control oocytes and 20 RPM oocytes) (** *p* < 0.01, * *p* < 0.05 vs. controls). (**b**) The graph shows that all the RPM oocytes analyzed display this phenotype. (**c**,**d**) The comparison of cumulative M-SER and MV distribution by the Kolmogorov–Smirnov test indicates that the two distributions are respectively significantly different. Dmax M-SER = 0.9 (red line); Dmax MV = 1(red line). The critical distance (Dcrit M-SER = Dcrit MV = 0.59), calculated at the 0.05 level, by Kolmogorov–Smirnov test, is lower than Dmax highlighting that the two distributions are significatively different. *p* value M-SER = 3 × 10^−4^; *p* value MV = 4 × 10^−5^.

**Figure 4 cells-12-01346-f004:**
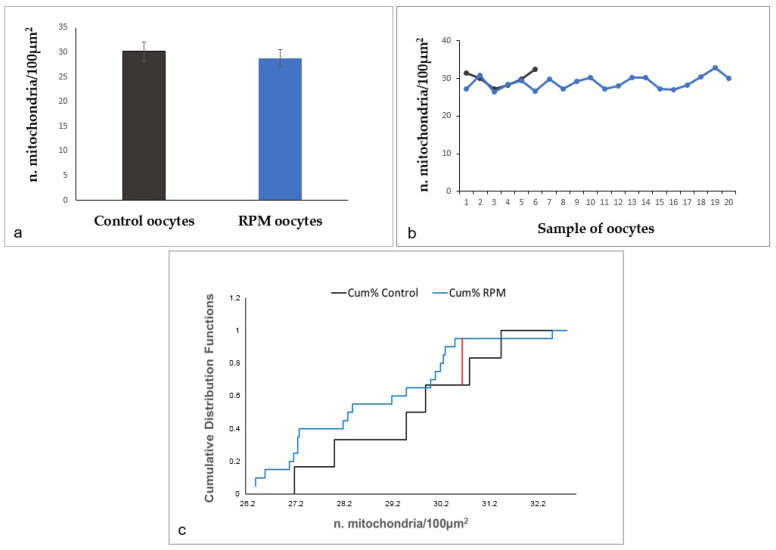
Morphometric evaluation of the number of mitochondria in human MII oocytes. (**a**,**b**) Microgravity does not affect the number of these organelles. Values are mean ± SD of the number of mitochondria/100 µm^2^ of oocyte area. There were no statistically significant differences (*p* = 0.29). (**c**) The comparison of cumulative mitochondrial distribution by the Kolmogorov–Smirnov test indicates that the two distributions are not significantly different. Dmax = 0.28 (red line). Dcrit = 0.58 calculated at the 0.05 level, by Kolmogorov–Smirnov test. *p* value = 8 × 10^−1^.

**Figure 5 cells-12-01346-f005:**
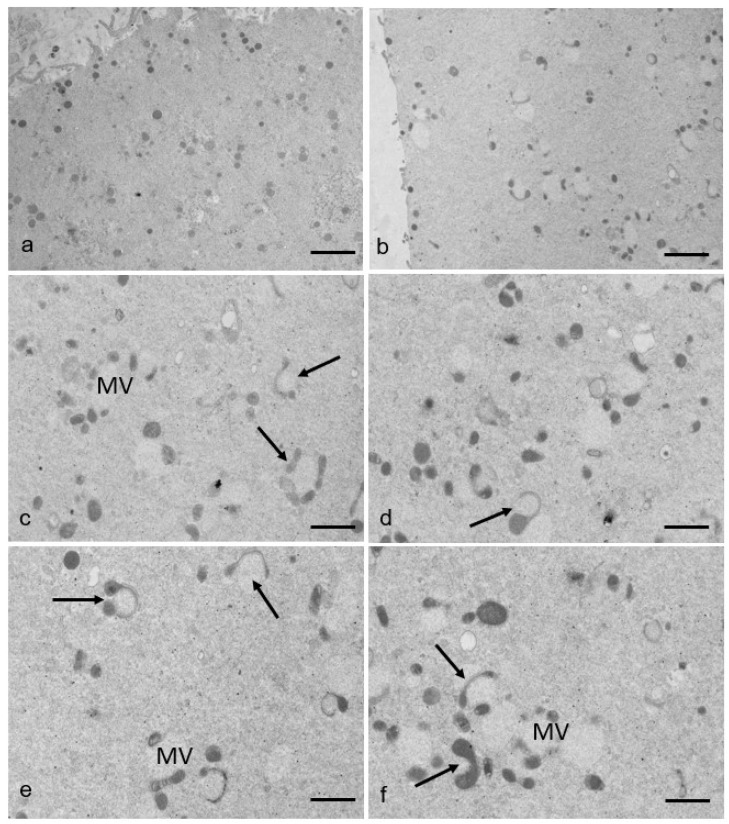
Mitochondrial fission in human MII oocytes. (**a**) In control oocytes, mitochondria display a rounded or oval shape with normal internal architecture. (**b**–**f**) In RPM oocytes, mitochondria seem affected by simulated microgravity, showing fission events (arrows). Bar = 1.6 µm (**a**,**b**). Bar = 1 µm (**c**,**d**); Bar = 800 nm (**e**,**f**).

**Figure 6 cells-12-01346-f006:**
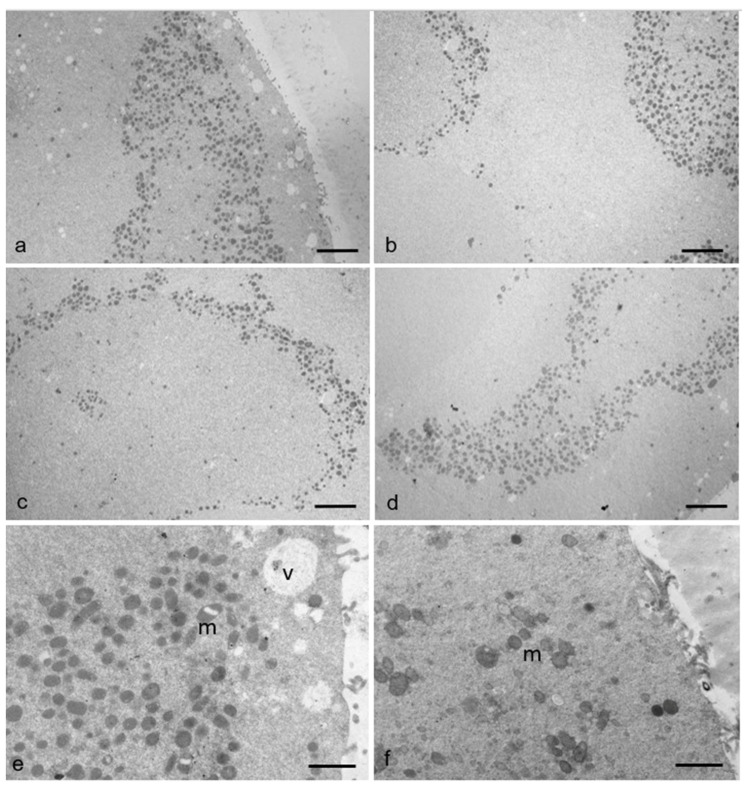
Mitochondrial distribution in human MII oocytes. (**a**–**e**) In some RPM oocytes, mitochondria (m) are clustered at the periphery of the cortical ooplasm. (**e**) Several vacuoles (v) in the ooplasm and inside mitochondria also occur in the cortical area suggesting possible structural damage. (**f**) In control oocytes, mitochondria show a diffuse localization in the ooplasm. Bar = 2 µm (**a**–**d**). Bar = 1 µm (**e**,**f**).

**Figure 7 cells-12-01346-f007:**
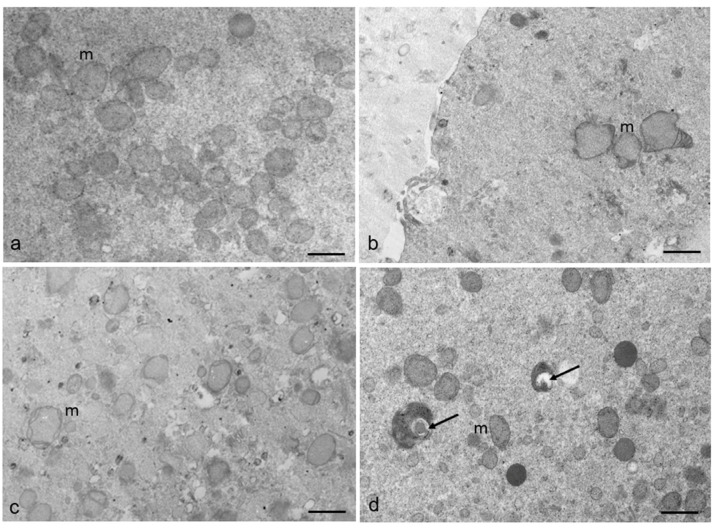
Mitochondrial morphology in human MII oocytes. (**a**) Mitochondria of control oocytes appear with a round and regular shape. (**b**–**d**). Note the presence of some irregular, swollen mitochondria (m) in RPM oocytes. The presence of large clear vacuoles or debris (arrows) inside the mitochondrial matrix also occurs in RPM oocytes. Bar = 800 nm.

**Figure 8 cells-12-01346-f008:**
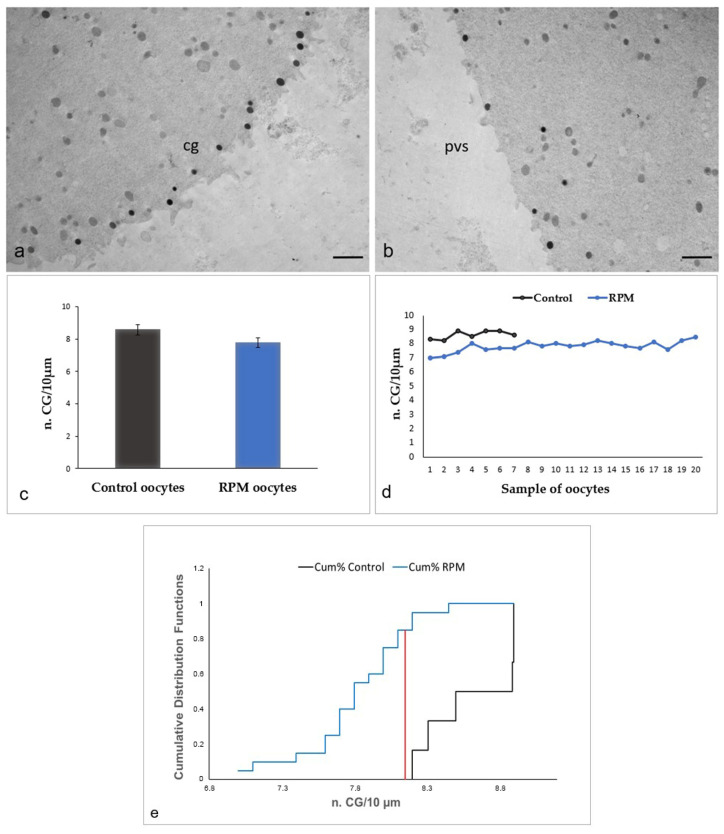
CGs in human MII oocytes. (**a**) In control oocytes, cortical granules (cg) are aligned slightly beneath the oolemma. (**b**) In RPM oocytes, peripheral CGs appear reduced, forming a discontinuous layer beneath the oolemma, whereas some appear relocated in the inner ooplasm. An enlargement of perivitelline space (pvs) may occur in RPM oocytes. Bar = 1 µm. (**c**,**d**) Morphometric evaluation of CG number in the two populations of oocytes (6 control oocytes and 20 RPM oocytes). (**c**) The graph indicates that microgravity does not significantly affect the number of suboolemmal CGs. Values are mean ± SD of the number of CG per 10 µm linear surface profile. There were no statistically significant differences (*p* = 0.06). (**d**) The graph shows that all the RPM oocytes analyzed display this phenotype. (**e**) The comparison of cumulative CG distribution by the Kolmogorov–Smirnov test indicates that the two distributions are significantly different. Dmax = 0.85 (line red). Dcrit = 0.58 calculated at the 0.05 level, by Kolmogorov–Smirnov test. *p* Value = 8 × 10^-4^.

**Figure 9 cells-12-01346-f009:**
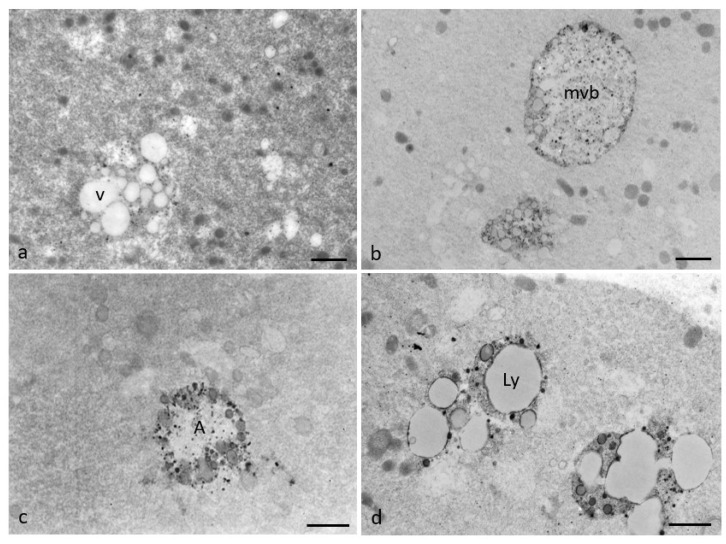
Representative pictures of RPM human MII oocytes by TEM. Microgravity induces ultrastructural cytoplasmatic alterations associated with cellular regression and degeneration. (**a**) Vacuoles (v); (**b**) multivesicular body (mvb); (**c**) autophagosome (A); (**d**) autophagolysosome (Ly). Bar = 1 µm (**a**,**b**); Bar = 800 nm (**c**,**d**).

**Table 1 cells-12-01346-t001:** Morphometric comparison of organelles distribution in human control and RPM oocytes. Data are mean ± SD. M-SER aggregates and MV complexes in RPM oocytes are statistically significantly different from control oocytes.

Morphometric Parameters	Control Oocytes	RPM Oocytes	*p*-Values of Significance
	Mean ± SD	Mean ± SD	
n. M-SER/100 µm^2^	1.97 ± 0.16	1.18 ± 0.31	*p* < 0.05
n. MV/100 µm^2^	3.58 ± 0.17	5.61 ± 0.52	*p* < 0.01
n. mitochondria/100 µm^2^	29.88 ± 1.94	28.86 ± 1.73	*p* = 0.29
n. CG/10 µm	8.61 ± 0.29	7.80 ± 0.35	*p* = 0.06

## Data Availability

Not applicable.

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
