# Peer review of "Effects of Simulated Microgravity In Vitro on Human Metaphase II Oocytes: An Electron Microscopy-Based Study"

_cells, 2023, doi:10.3390/cells12101346_

Round 1
Reviewer 1 Report
In this study, the authors use an established method for generating microgravity or weightlessness oocytes using an RPM machine. The authors collect 26 human oocytes and test for various parameters of oocyte quality. The authors find differences in the RPM subjected oocytes versus the control in terms of mitochondrial features and and aggregation as well as some other features in the cell. As of now, the paper is descriptive and cannot reflect on whether these changes have functional consequences. The presentation of the data is not clear and labeled well (in comparison to the discussion which seemed long and protracted) and it is also unclear whether these results are specific to oocytes versus just mitochondria in any cell type. From the discussion it appears that the phenotypes noted may be present in most cells under microgravity, in which case the functional significance of this study is incremental.
Here are some major comments:
1. The human oocytes were split quite randomly and in a skewed way in 20 and 6 groups. Is there a reason for this? Did the authors anticipate any loss of oocytes during RPM exposure? This skew makes any tests of significance very hard to assess.
2. I am assuming that the human oocytes picked were ones that were not chosen for IVF/ICSI cycles? So they are clearly the ones that were discarded?
3. In each figure it is unclear how many out of the 20 and 6 oocytes had the phenotype in question. This must be changed for this paper to represent a quantitative assay and to come up to recent basic standards of reporting. For e.g. figure 3 should show all data points as circles (should be 20 in RPM and 6 in WT) and then we can see if there is a variation and how consistent the data is. That should be done for all the figures, bar graphs are really not showing all the data.
4. Another main concern is that the figures 4-7 do not appear to have control TEM images? That does not allow for any comparison to be had at all. Control images must be added or at least shown for every TEM figure. Without it, we cannot evaluate the data. The controls in figure 2 cannot serve as controls for figure 4-7. Are there no other images? Figure 2 says note that mitochondria appear rounded or oval (line 250) but these are not marked at all! Another major concern is that the remaining figures are qualitative with no quantification except for in one case of the CGs, which should again be presented as data points. How frequently do they see this happening in 20 oocytes compared to controls? Is it more than what is reported for the soma or confirms those results?
5. Lastly, it occurs to me that human oocytes are hard to get. But surely mouse oocytes would not react that differently and would represent a more robust source of material to be able to get more quantitative and informative data that can then be correlated with the data from the human oocytes. Functional assays of whether the oocytes are competent or not, which is the significance of this project can be performed in mice, whereas here we do not know if what we are seeing is actually affecting IVF or other reproductive capacity.
Minor point that the scale bars on figure 4 by eye appear to be 1 um for cdef and 800 for a,b. Not what the authors have in the figure legend.
Author Response
Please, look at the attached file

Reviewer 2 Report
In this manuscript, the authors sought to examine the impact of microgravity to human MII oocytes. They identified changes in ooplasmic features and mitochondria morphology. This manuscript is interesting, relevant to the field and will provide better understanding of how space environment could affect the body systems of human beings. However, the thoroughness of the manuscript needs to be further improved to be able to make the conclusions clear. General comments are as below:
1. many of the conclusions, especially throes regarding to mitochondria behaviour and activities are lacking quantifications. For example, what's the frequency of mitochondria fission in RPM vs control oocytes? In terms of the clustering phenotype, the authors should provide further quantifications on the size/ number of mitochondria in a cluster/distance between mitochondria.
2. the authors should consider providing immunostaining for cytoskeleton components key to maintain cellular structures, including mitochondria, polarity complex (Par6/aPKC) and actomyosin complex. The actomyosin complex has been very well described for its importance to maintain the oocyte morphology, alterations in its activity/localisation cannot be revealed simply by TEM.
Author Response
Please, look at the attached file

Round 2
Reviewer 1 Report
Article is much improved. But it still remains qualitative.
As the authors suggest future experiments are required to understand the full implications of these findings on actual fertility outcomes and hence may be suitable for a less mechanistic journal but I leave that decision up to the editors.
Author Response
Please, see the attachment

Reviewer 2 Report
the authors have sufficiently addressed my concerns.
